# Trephination versus Minimally Invasive Transnasal Approaches for the Diagnosis and Treatment of Sinus Disease in Horses

**DOI:** 10.3390/vetsci9070334

**Published:** 2022-07-01

**Authors:** Matthias C. Jehle, Nora M. Biermann, Eva Haltmayer

**Affiliations:** Equine Hospital, University of Veterinary Medicine Vienna, Veterinärplatz 1, 1220 Wien, Austria; nora.biermann@vetmeduni.ac.at (N.M.B.); eva.haltmayer@vetmeduni.ac.at (E.H.)

**Keywords:** sinusitis, equine, minimally invasive, electrosurgery, laser surgery, balloon sinoplasty

## Abstract

**Simple Summary:**

The horse’s upper respiratory tract contains several sinus cavities, and the infection of these sinuses (sinusitis) can be encountered due to various reasons. Several different diagnostic and treatment options have been descripted in the last few decades to facilitate prompt and correct diagnosis and treatment. With advances in the development of several diagnostic tools, for example three-dimensional imaging (e.g., standing computer tomography) and small-diameter video endoscopes, new therapeutic techniques are constantly being introduced. This report aims to highlight the advantages and disadvantages of several invasive and minimally invasive techniques to enter the affected sinuses in a retrospective clinical study. Nearly half of the cases presented here could be entered and subsequently treated through the nose by the use of a small-diameter video endoscope without additional intervention. However, in cases where surgery was required, minimally invasive techniques seemed to be as effective in resolving clinical signs as more invasive approaches. While proper case selection is crucial for success, these experiences point towards the fact that minimally invasive transnasal approaches should be considered more commonly for the diagnosis and treatment of horses presenting with sinusitis in the future.

**Abstract:**

This retrospective case series aims to highlight the advantages and disadvantages of different treatment options for sinus disease in horses, with a special focus on surgical approaches. The medical records of 25 horses with sinusitis which presented to the equine hospital, Vetmeduni Vienna, between August 2020 and January 2022 were analysed. In 11 cases, conservative treatment by the transnasal endoscopic lavage of the affected sinuses through the sinonasal channel or through a developed, pre-existing sinonasal fistula was performed. In the remaining 14 cases, openings into the sinus for subsequent lavage were surgically created either via classical trephination/bone flaps (8) or by transnasal endoscopically guided minimally invasive approaches (8). In some cases (2), a combination of classical extra-nasal and minimally invasive transnasal approaches was required. The minimally invasive techniques used were comprised of laser surgery (3), electrosurgery (3) and balloon sinoplasty (2). The sinusitis and underlying diseases were successfully treated in all of the horses until hospital discharge. Long-term follow-up could be obtained only from a small number of patients due to the temporal proximity (the study began two years ago) to their initial discharge from the hospital, but they showed the continuous resolution of clinical signs in all cases in which follow up data were available. The results of this case series indicate that if surgical intervention is required, minimally invasive techniques appear to be a promising and potentially cheaper alternative to classical extra-nasal approaches for the treatment of sinus disease in selected cases. In the future, the use of these techniques should be considered more commonly with the broader availability of improved endoscopes and instruments. Additionally, by using small-diameter endoscopes, the need for any surgical intervention might be significantly reduced in many cases.

## 1. Introduction

Sinusitis—an infection of one or more sinuses—is a relatively rare disease in horses and can be challenging to diagnose and treat due to the complex anatomy of the equine sinus system. Additionally to the large sinus systems, in 20% of cases the neighbouring bullae of the dorsal and ventral conchae are also affected [1].

In general, sinusitis is classified by cause into primary and secondary forms. In cases of primary sinusitis, which comprise approximately 25–35% of all cases of sinusitis, the inflammation of the mucosal lining of the sinuses, most likely due to prior upper airway infection, results in the accumulation of purulent discharge within the sinus system without any other apparent cause [2,3]. All other cases of sinusitis are considered secondary to an underlying pathology, and are comprised of dental disease (20–24% of cases), sinus cysts (13–14% of cases), trauma (6.1–6.5% of cases), neoplasia (5–7.9% of cases), and ethmoid haematoma (3.5–7.6% of cases), as well as sinus mycoses (3.5–4.7% of cases) [2,3].

Most frequently, affected horses present with one-sided nasal discharge of varying types (88%). Other frequent clinical signs are swellings of the facial and maxillary bones (46%), inflammatory reactions of the mandibular lymph nodes (>50%), and malodorous nasal discharge, especially in cases of dental sinusitis [2].

Diagnosis is achieved through a combination of a thorough clinical and dental examination, a transnasal endoscopy of both nasal passages and the upper airway, including the guttural pouches and head radiographs, and in complicated cases computer tomography (CT) of the head.

Endoscopic changes of the nasal passages or sinus systems, such as discharge from the drainage angle or mucosal swelling, have been reported as being present in 91% of cases, although endoscopy only contributes to a precise diagnosis in 21% of cases [2]. For the endoscopic examination of the middle nasal meatus and the area of the sinonasal apertures, endoscopes with a smaller diameter (less then 10 mm) are recommended [4]. In chronic cases of sinusitis, enlarged sinonasal openings might already have developed due to the necrosis of the conchal bones. In these cases, direct sinuscopy through these openings is often possible [3].

Radiographic changes of normal sinus appearance, such as increases in the soft tissue density of affected sinuses, are visible in 81–89.5% of cases but lead to a final diagnosis in only 36% of cases, mainly due to the large number of overlying structures at the level of the equine skull [2,5]. Head CT has a 100% sensitivity and 96.7% specificity [5], and should be considered in complicated cases despite its higher costs and the potential need for general anaesthesia [6].

Most cases of primary sinusitis and practically all cases of secondary sinusitis have been reported to require some kind of surgical intervention [3]. Dixon et al. reported that they treated 21% of acute primary sinusitis successfully conservatively; however, this did include the lavage of the affected sinuses through a small trephination [3]. Nevertheless, the overall prognosis is good to excellent, with 91% of cases recovering fully and 7% of cases recovering at least partially in primary sinusitis, and 82% recovering fully in secondary dental sinusitis [3]. It is important to keep in mind that, in cases of secondary sinusitis, the treatment of the underlying cause is mandatory for long-term resolution of clinical sings [3]. Classical extra-nasal surgical techniques, including maxillary and frontal trephinations and bone flaps, have successfully been used to gain access to the sinus system and treat sinusitis [3,7,8,9,10,11,12,13] with or without sinuscopy via the externally created openings [3,8,10,14,15,16,17]. However, with improvements in the quality and size of endoscopes and instruments, transnasal endoscopically guided techniques have also been successfully used in the last 20 years [4,12,18,19,20,21,22,23]. These minimally invasive transnasal techniques can be divided into three groups. First, electrocautery can be used to form openings in the dorsal and ventral conchae, and thereby gain endoscopic access to the underlying sinuses and recesses [4,12,22]. Second, transendoscopic laser surgery, mainly facilitated by the use of highly flexible diode lasers are applied to fenestrate the dorsal and ventral conchae, dorsal and ventral bullae, the septum between the ventral conchal recess and sinus as well as the palatine bone [4,17,18,20,21,22,23]. This technique can also be used within the sinuses. The third technique described here, is the widening of the nasomaxillary aperture with a balloon catheter placed under endoscopic guidance, similar to balloon sinuplasty in human medicine [19]. All three minimally invasive techniques, as well as classical surgical approaches have been used and compared in this case series.

## 2. Materials and Methods

Medical Records from August 2020 to January 2022 from all horses admitted for unilateral, purulent nasal discharge were reviewed for the presence of sinus disease, the type of diagnostic method, and the methods of treatment: trephination vs. minimally invasive techniques. Regarding the treatment of the animals at the time, the owners consented to the treatment of their animals in a standardized admission form, and signed an informed consent form. Because all of the horses presented in this study underwent routine therapy for sinusitis, no additional approval from the institutional ethics board was needed for this paper.

### 2.1. Diagnostic Work up

In all cases, a clinical examination and routine blood work (CBC, chemistry) was performed at the time of admission.

For all of the standing procedures, the patients were restrained in stocks, and sedated with a combination of butorphanol 0.01 mg/kg IV (Butomidor, Richter Pharma AG, Wels, Austria) and detomidine 0.01 mg/kg, IV (Equidor, Richter Pharma AG, Wels, Austria). A full dental examination under standing sedation, using a video endoscope, was performed. Dental pathologies were recorded, and a routine odontoplasty was performed where necessary. Additionally, in all of the horses, a radiographic examination of the head was performed. The radiographic views comprised at least lateral–lateral, dorsoventral and oblique views of the upper dental arcades and paranasal sinuses. If it was deemed necessary by the clinician in charge, a CT of the head was performed under general anaesthesia.

### 2.2. Diagnostic and Therapeutic Procedures

#### 2.2.1. Transnasal Sinus Endoscopy (TNSE)

In order to improve compliance with the procedure, Mepivacaine 2% (Mepinaest; Gebro Pharma GmbH, Fieberbrunn, Austria) was applied topically via the working channel of the video endoscope within the nasal cavity and on the conchal mucosa, as needed.

In the present study, TNSE was used as a diagnostic and a therapeutic tool.

For TNSE and all of the transnasal endoscopic procedures, a 5.9 mm paediatric gastroscope of 1 m length (60 511 PKS/NKS; Karl Storz, Tuttlingen, Germany) was used. A full evaluation of the upper airways, including the guttural pouches, larynx and pharynx was performed. The nasal cavity was evaluated for mucosal oedema, purulent discharge, mucosal masses, swellings and deformities of the nasal conchae and the ethmoid turbinates. From the middle nasal meatus, it was attempted to gain access to the rostral and caudal sinus system over the sinonasal channel via the rostral and/or caudal nasomaxillary aperture.

When severe mucosal swelling was present on initial examination, Flunixin 1.1 mg/kg (Niglumine, Laboratorios Calier S.A., Barcelona, Spain) was administered systemically twice daily for 3 to 5 days in order to reduce the mucosal oedema and thereby improve access to the sinus system via the nasomaxillary passage.

In cases were the natural opening of the nasomaxillary aperture was too narrow for the video endoscope to pass and the administration of flunixin did not improve access, the following approaches were used to gain access to the affected sinus. Figure 1 provides an overview for the minimally invasive endoscopic approaches and should be seen as a supplement of the other figures provided.

#### 2.2.2. General Preparation

All of the invasive procedures (e.g., conchotomies, laser fenestration, SENMAP) were performed under standing sedation and regional aesthesia. A maxillary nerve block was performed on the respective side under aseptic conditions with 10–15 mL of Mepivacaine 2% (Mepinaest; Gebro Pharma GmbH, Fieberbrunn, Austria) using the Extraperiorbital fat body insertion technique (EPFBI technique) [24].

The minimally invasive procedures were commonly performed under standing sedation, with the local application of Mepivacaine 2% through the working channel of the video endoscope as needed.

#### 2.2.3. Widening of the Nasomaxillary Opening/Channel Using a Foley Catheter

The widening of the nasomaxillary openings was performed in accordance with the bulloplasty procedure described by Bell et al., 2009 [19]. A 1.2 mm stainless steel cerclage wire was used as a guide wire for a 40 cm, 20CH balloon catheter (Rüsch Gold Ballonkatheter, Rüsch Austria GesmbH, Vienna, Austria). The catheter was placed in the rostral or caudal nasomaxillary aperture, under endoscopic guidance (Figure 2). If necessary, the catheter was placed using flexible endoscopic grasping forceps. Subsequently, the balloon of the catheter was inflated using 15 mL Ringer’s solution and was either left in place for 48 h or removed right after the procedure, with the goal to enlarge the respective aperture enough to pass an endoscope through it. Transendoskopic sinus lavage to remove purulent material was performed once access was gained.

#### 2.2.4. Endoscopic Guided Conchotomy

In order to gain access to the rostral or caudal sinus system, a conchotomy of the ventral or dorsal conchae was performed as described by Bach et al., 2019 [12].

In brief, an electrosurgical conchotome (450 mm long, 6 mm circular steel shafts with a 5 mm diameter ball tip), set at 80 W for coagulation and 50 W for cutting, was inserted into the middle nasal meatus and used to create a full-thickness defect of approximately 50 mm in the medial conchal wall. The landmarks for the conchotomy of the ventral conchae were 10–50 mm rostral to the caudal end and 15–20 mm dorsal to the ventral border of the ventral conchae (Figure 3) [12].

In the current study, the landmarks for the conchotomy of the dorsal conchae were 10–50 mm rostral to the caudal end and 10–20 mm dorsal to the ventral border of the dorsal conchae (Figure 4 and Figure 5).

#### 2.2.5. Transendoscopic Laser Fenestration of the Nasal Conchae

The laser fenestration of the medial wall of the ventral or dorsal conchae was performed similarly to the procedure described by Morello et al., 2010. [20] The main difference was that in the current study, the endoscope was positioned in the middle nasal meatus and the laser fiber was passed through the working channel of the endoscope. The landmarks to access the ventral or dorsal conchae were used as described in Section 2.2.4.

A full-thickness defect was created in the conchal wall using a 810 nm diode laser (MedArt 426, Asah Medico A/S, Hvidovre, Denmark) with a 600 μm laser fiber set to the 18 W pulsed mode.

#### 2.2.6. Transendoscopic Fenestration of the Bulla of the Maxillary Septum (BMS)

In accordance with the described conchotomy in Section 2.2.4., the video endoscope was positioned in the middle nasal meatus on top of the ventral conchae, and advanced to the caudal nasomaxillary aperture, such that the endoscope was eventually placed on top of the bulla of the maxillary septum (BMS). The BMS was subsequently fenestrated at a length of approximately 10–20 mm using the diode laser as described in Section 2.2.5 (Figure 6, Figure 7, Figure 8 and Figure 9).

#### 2.2.7. Trephination, Surgical Enlargement of the nasomaxillary aperture (SENMAP) and Frontonasal Boneflap Procedure

In this study, the SENMAP technique described by Bach et al., 2019 [13] was either performed when minimally invasive techniques were not successful in accessing the sinus or when the underlying sinus pathology required more extensive access to the respective sinussystem, e.g., mass or cyst removal.

In brief, a conchofrontal sinus trephination was performed centred on the intersection of a line that was parallel to the long axis of the head and 60% of the distance from the midline to the medial canthus of the eye and a line 2.5 cm caudal to a line joining both medial canthi. After the resection of the ventral lamella of the dorsal conchae and the dorsal aspects of the BMS using electrosurgical instruments (80 W, coagulation mode), a fenestration (maximum 5 cm × 5 cm) was made in the medial wall of the ventral conchae using a bipolar vessel sealing device (MarSeal Slim IQ; KLS Martin, Tuttlingen, Germany) to gain access to the nasal cavity [13].

In one case, a frontonasal boneflap—as described by Schumacher et al., 2000 [7]—was created due to the size and the rostral extent of the sinus cysts. Once access to the sinus was achieved, the SENMAP procedure was performed as described above.

In cases of sinus cysts, the removal of the cystic tissues resulted, in some cases, in openings into the nasal passage due to previous pressure necrosis of the conchal structures, and negated the need to create an opening by the bipolar vessel sealing device.

Once the access into the nasal cavity was established, the trephination site was closed. Only the skin was sutured with a single interrupted vertical mattress pattern using USP 1 polypropylene (Optilene, B. Braun Austria Ges. m.b.H., Maria Enzersdorf, Austria).

All of the subsequent treatments were performed using transnasal approaches.

### 2.3. Additional Therapeutic Procedures

#### 2.3.1. Sinus Lavage

After the affected structure was accessed, purulent material was removed via lavage with sterile Ringer’s solution over the working channel of the video endoscope. Lavages were performed in regular intervals (every 2–4 days) until signs of sinus disease subsided.

For the sinus lavage, a commercially available laparoscopy pump (Laparomat 263220 20, Karl Storz SE & Co. KG, Tuttlingen, Germany) was connected to the working channel of the video endoscope, set at a pressure of 600 mmHg and a flow rate of 1600 mL/min. The lavage volumes varied greatly between cases depending on the amount and ease of removal of purulent material, but ranged between 2 and 6 L per lavage.

#### 2.3.2. Exodontia

In cases were dental pathologies such as apical infection and endodontic or periodontal disease could be identified as the causative agent of the sinusitis, exodontia of the affected tooth/teeth was performed in addition to the sinus lavages. The exodontia was performed under standing sedation and regional anaesthesia (as described in Section 2.2.2).

#### 2.3.3. Removal of the Sinus Cysts

The removal of the sinus cysts was performed under standing sedation and regional anaesthesia, either via the trephination/SENMAP technique or via the above described minimally invasive approaches. In most cases in this study, a combination of approaches was used to remove the cystic tissue. If necessary, the transnasal resection of the cyst under endoscopic guidance using a bipolar vessel sealing device inserted into the middle nasal meatus was performed via the repeated application of the device at the base of the cyst.

## 3. Results

A total of 25 horses were included in the current retrospective study.

A detailed description of the patient signalment, diagnoses and procedure performed is displayed in Table 1.

### 3.1. Diagnostic and Therapeutic Procedures

#### 3.1.1. TNSE

In 11 cases, access to the affected paranasal sinus was possible via the natural openings of the sinonasal channel. In five of these cases, access was facilitated during the course of disease, due to ventral conchal necrosis and the natural development of sinonasal openings. In four of these 11 cases, access to the sinus via the nasomaxillary aperture was only possible after the systemic administration of Flunixin-Meglumine (1.1 mg/kg, SID, PO) for a mean duration of 4 days. Repeated lavages of the affected paranasal sinus were performed a mean frequency of 4.5 times until all of the purulent material was cleared from the sinus and signs of sinusitis had significantly improved (*n* = 4) or had completely resolved (*n* = 7). In the remaining 14 cases, either minimally invasive procedures (8 horses) or a trephination/SENMAP technique were used to access the affected paranasal sinus. In two horses, both a minimally invasive technique and trephinations was performed. In only one case, due to the extensive nature of the sinus cysts, a decision for a frontonasal bone flap was made to improve access.

#### 3.1.2. Balloon Sinuplasty

A balloon sinuplasty was performed in two cases. In one of these cases, however, the balloon catheter had to be placed in a retrograde fashion through an open fracture of the maxillary bone (Figure 10 and left in situ for 48 h due to the severe swelling of the nasomaxillary aperture.

#### 3.1.3. Conchotomy and Transendoscopic Laser Fenestration of the Bulla of the Maxillary Septum

Laser fenestration of the BMS was used in three cases and also in three cases a conchotomy of the medial wall of the ventral (*n* = 2) or dorsal (*n* = 1) conchae was performed to access the paranasal sinus.

In one of these cases the conchotomy was unsuccessful in providing access to the rostral maxillary sinus due to the long reserve crows of 207 and 208 impairing passage from the ventral conchal sinus over the infraorbital canal into to rostral maxillary sinus.

#### 3.1.4. Trephination and SENMAP

Of the eight horses that had a trephination/SENMAP procedure and/or a bone flap performed, three had sinus cysts and one an ethmoid hematoma removed during surgery. In all cases, access into the nasal cavity was established in the area of the medial wall of the ventral and/or dorsal conchae.

In two cases, bilateral masses (cysts, ethmoid hematomas) were diagnosed that were located within the rostral nasal cavity in the area of the ventral conchal recess. These masses were removed via laser resection and/or resection with a bipolar vessel-sealing device trans-nasally.

A mean of 7 sinus lavages were performed after surgical access to the paranasal sinus (the minimally invasive procedure and trephinations). In two cases with multiple and more complex pathologies, sinus lavages were performed 17 and 29 times, respectively.

All of the surgically created openings into the sinus system (the minimally invasive procedure and SENMAP) were still wide and patent at the time of hospital discharge.

A follow up endoscopic examination after hospital discharge (via endoscopic examinations) was recorded in five cases due to the limited time span after treatment commenced for the enrolled cases.

In two cases, the SENMAPs were widely open and patent 6 weeks after surgery, and in a third case 10 month after surgery. No signs of sinusitis or the recurrence of cysts were found in these cases on re-examination. In another case, the SENMAP opening in the medial wall of the ventral conchae was closed to a slit-like opening due to adhesions after 5 months, and purulent discharge was draining out of this opening. In the fifth case, the opening of the performed conchotomy of the ventral conchae was slightly smaller but still patent at 6 months after surgery; however, signs of sinusitis were still present due to endodontic disease of the left first molar tooth (209).

### 3.2. Complications

Haemorrhage was the most frequent complication with the minimally invasive and SENMAP techniques in the current study. After all of the SENMAP procedures (eight horses) as well as after two conchotomies and the frontonasal bone flap, significant haemorrhage occurred and the sinus and/or nasal cavity had to be packed with sterile gauze. The gauze could be removed after 2 days in all cases. No further bleeding was observed.

During one of the laser bullotomies and during one conchotomy, mild haemorrhage impaired visualisation. Therefore, the procedure was aborted, but bleeding subsided without further treatment. Access to the affected sinus was possible 2–3 days later due to the effect of thermal necrosis of the treated tissues.

Another frequent complication seen with cases of chronic sinusitis was the necrosis of the ventral conchae (five horses) in the course of the disease. However, these sinonasal openings facilitated access to the sinus system and improved drainage.

Other complications in this study were suture periostitis (*n* = 1), the fracture of the nasolacrimal duct (*n* = 1), and a surgical-site infection of the frontonasal bone flap (*n* = 1) (Table 1).

## 4. Discussion

Previously published articles and the results presented in this current study show that with careful selection of suitable cases good to excellent results via minimally invasive transnasal methods can be achieved [4,12,22]. Complications, for example severe haemorrhage, suture dehiscence, wound infections and suturitis described in extranasal surgical approaches can mostly be avoided using minimally invasive techniques [8,25,26,27]. Additionally, these approaches have an excellent cosmetic outcome, usually reduced duration of surgery and have the potential to be performed as an outpatient procedure. Resulting costs might often be less compared to classical surgical approaches. Therefore, minimally invasive procedures increase owner satisfaction as well as animal well-being and can reduce time to hospital discharge compared to classical approaches.

In chronic cases of sinusitis, the natural openings between the nasal cavity and the paranasal sinus can be enlarged, which allows direct transnasal access with a 5–7 mm video endoscope. In our case series, no surgical access to the affected sinus system was required in nearly half (44%) of the cases. In these cases, the transendoscopic lavage of the affected paranasal sinus was possible through the natural openings or through openings formed as sequelae of ventral conchal necrosis. Even if a passage through the nasomaxillary aperture was not possible at the initial examination, in several cases, the administration of NSAIDs for several days lead to a reduction of mucosal oedema and eventually provided transnasal access to the affected sinus system. In addition, if the underlying cause of sinusitis is related to a tooth pathology, exodontia of the affected tooth/teeth might also improve mucosal swelling. Therefore, based on our experience, an initial conservative approach before performing invasive procedures may be considered in selected cases.

Balloon sinuplasty was used in cases where the BMS could not be accessed with the laser fibre, or when access to the ventral conchal sinus was not needed. In both cases, sufficient transnasal endoscopic accessibility to the sinus was achieved over the indentation left in the BMS by the balloon. There have been concerns in the literature that creating an indentation in the BMS may cause insufficient drainage from the ventral conchal sinus to the rostral maxillary sinus [12]. To date, we have not encountered this previously described complication in our cases, but further research and larger case numbers are certainly required for more reliable results.

The transendoscopic fenestration of the BMS via laser surgery is a technique which, to our knowledge, was first described here. It allowed sufficient access to the rostral and the caudal sinus system, and substantially aided therapeutic success in cases where it was performed. One big advantage of the transendoscopic fenestration is that access to both sinus systems, rostral and caudal, can be achieved with only one minimally invasive access. While in the authors experience the amount of haemorrhage encountered is usually only mild, it is still more severe compared to electro surgery.

Conchotomies have been successfully used to gain access to the respective sinus system for several years in our clinic. One limitation of electro-surgical conchotomies is the rigid design of the hand pieces that only allow safe application in the common nasal meatus. In our experience, the risk of collateral thermal trauma to the nasal septum is increased with this technique compared to the transendoscopic diode laser fenestration of the conchal wall, although no serious damage was experienced in the cases presented here. However, electro surgery provides superior haemostasis and larger-diameter openings into the sinus cavities. The authors therefore prefer the use of the electro conchotome, whenever adequate access to the area is possible.

Limitations of these minimally invasive procedures exist, particularly in younger horses (<6 years) when the rostral sinus system is involved. In these cases, long reserve crowns can potentially impair passage from the ventral conchal sinus over the infraorbital canal into the rostral maxillary sinus following a conchotomy of the ventral conchae, again requiring more invasive options for access.

Patients with sinus cysts and long-lasting chronic sinusitis might present a particular challenge to transnasal endoscopic approaches, as considerable changes of the sinus anatomy may be present, and may impair endoscopic passage.

Dentigerous cysts, cementoma, neoplasia and ethmoidal hematomas are usually not removable via minimally invasive transnasal approaches due to their size and location within the sinus system and or the nasal cavity. Sinus cysts can be opened and drained using an endoscopic approach, but the complete removal of the cyst is usually not possible. In these cases, a classical trephination or bone flap procedure would be the technique of choice to ensure complete cyst removal and therefore reduce the risk of recurrence [28]. On the other hand, inspissated purulent material (chondroids) can often be removed by repeated transnasal endoscopic lavages; however, this may take considerable time and patience.

The most relevant disadvantage of minimally invasive approaches, in the authors’ experience, is that—in contrast to more invasive techniques like SENMAP—endoscopically created openings often tend to quickly reduce in size, and might completely close during the healing process and thereby impair permanent drainage and endoscopic access. In the one case in this study where a follow up endoscopic examination was performed, access was still possible at the control examination 6 months after surgery. However, especially in cases of primary sinusitis once clinical signs have completely resolved, access to the respective sinus may no longer be required, but larger case numbers and longer follow up times are required to determine its relevance in the clinical setting.

Ultimately, in cases where permanent drainage and accessibility is required, for example in cases suffering from sinus mycosis or cases with impaired drainage over the natural openings, a classical trephination combined with a SENMAP procedure is recommended.

## 5. Conclusions

Minimally invasive endoscopic approaches to the sinus system and conservative treatment via transnasal endoscopy provide several advantages compared to classic extranasal approaches. Nevertheless, suitable cases should be selected carefully. Excellent anatomical knowledge and proper instrumentation are mandatory for successful treatment. Considering the continuous development and broader availability of small-size endoscopes and minimally invasive surgical instruments, minimally invasive techniques provide a promising alternative for the treatment of sinusitis in horses in the future.

## Figures and Tables

**Figure 1 vetsci-09-00334-f001:**
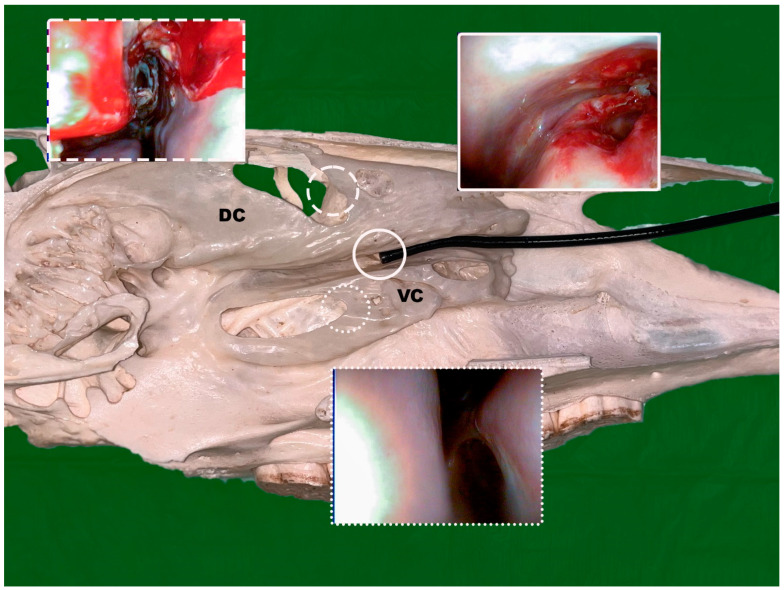
Axial view with the endoscope placed in the middle nasal meatus on top of the ventral conchae (VC) with the distal end of the endoscope int the area of the sinonasal channel; endoscopic views are shown after conchotomy of the dorsal conchae (DC) (dashed line), the ventral conchae (spotted line) and after laser laser fenestration of the bulla of the maxillary septum (solid line); (Photo: Matthias Jehle, Vetmeduni Vienna).

**Figure 2 vetsci-09-00334-f002:**
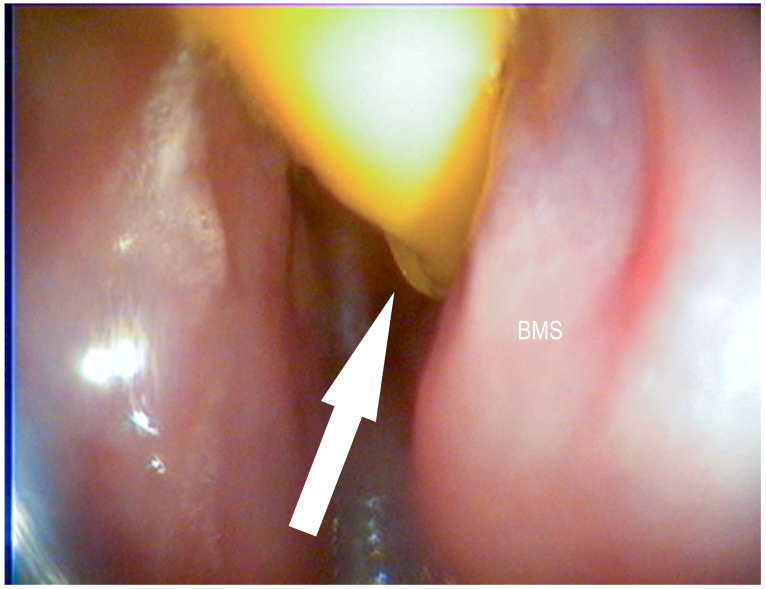
Foley catheter inserted into the medial nasal meatus and placed in the caudal sinonasal canal (white arrow) on top of the bulla of the maxillary septum (BMS) (Photo: Matthias Jehle, Vetmeduni Vienna).

**Figure 3 vetsci-09-00334-f003:**
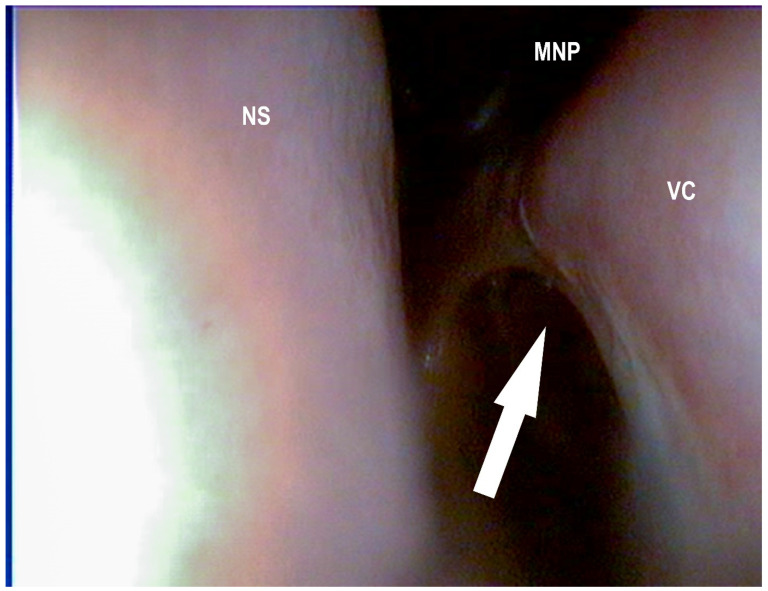
Surgically created opening to the ventral conchal sinus (white arrow) after electro surgical conchotomy: ventral conchae (VC), middle nasal passage (MNP), nasal septum (NS) (Photo: Matthias Jehle, Vetmeduni Vienna).

**Figure 4 vetsci-09-00334-f004:**
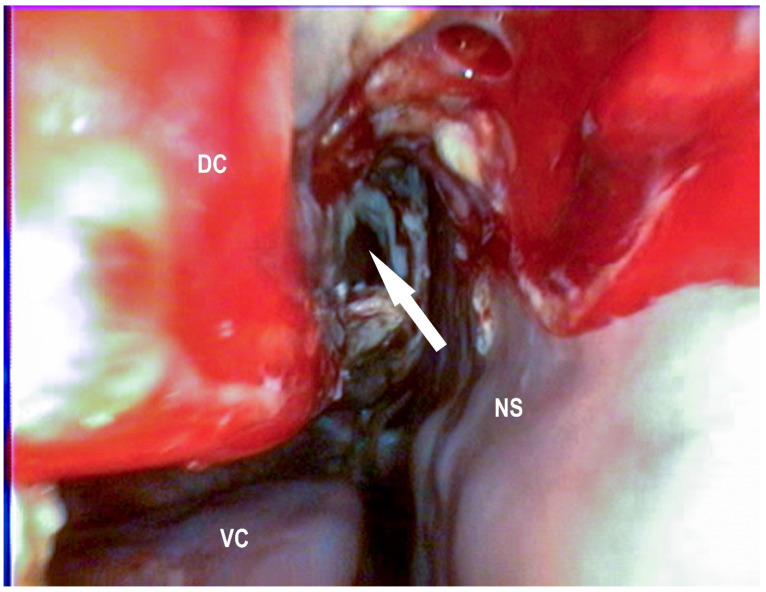
Surgically created opening to the dorsal conchal sinus after electro conchotomy two days after surgery (white arrow): ventral conchae (VC), dorsal conchae (DC), nasal septum (NS) (Photo: Matthias Jehle, Vetmeduni Vienna).

**Figure 5 vetsci-09-00334-f005:**
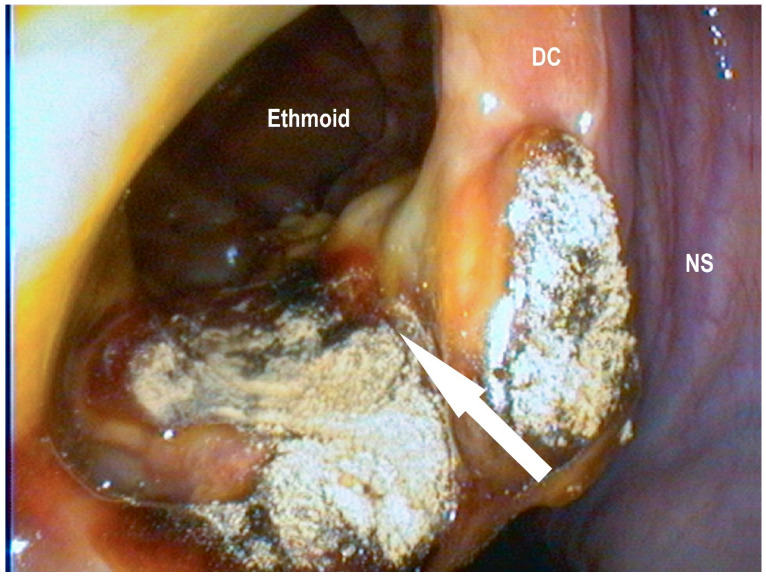
Conchotomy site of the same horse as in Figure 4, approximately 3 weeks after surgery with a view in the dorsal conchal sinus (white arrow), dorsal conchae (DC), and nasal septum (NS) (Photo: Matthias Jehle, Vetmeduni Vienna).

**Figure 6 vetsci-09-00334-f006:**
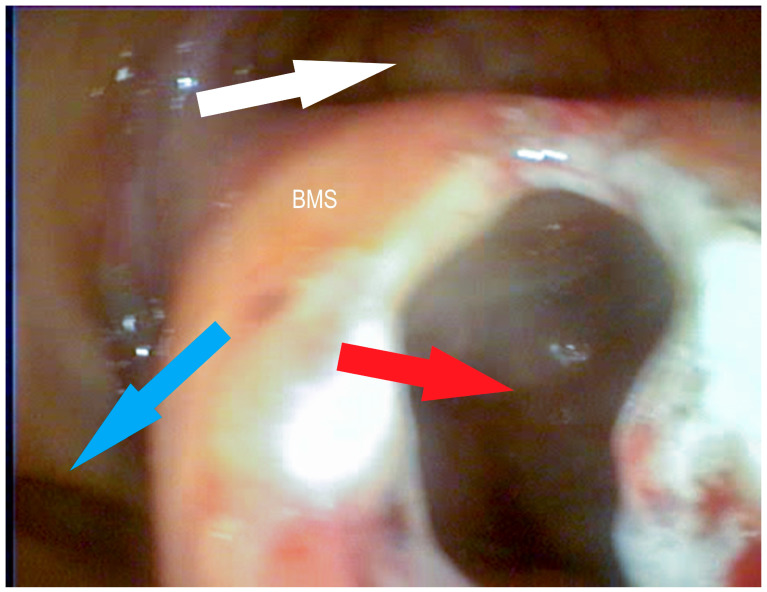
Bulla of the maxillary septum (BMS) after laserbullotomy; surgically created passage to the rostral maxillary sinus (red arrow); natural passage to the caudal maxillary sinus through the caudal sinonasal canal (white arrow) and to the common nasal meatus (blue arrow) (Photo: Matthias Jehle, Vetmeduni Vienna).

**Figure 7 vetsci-09-00334-f007:**
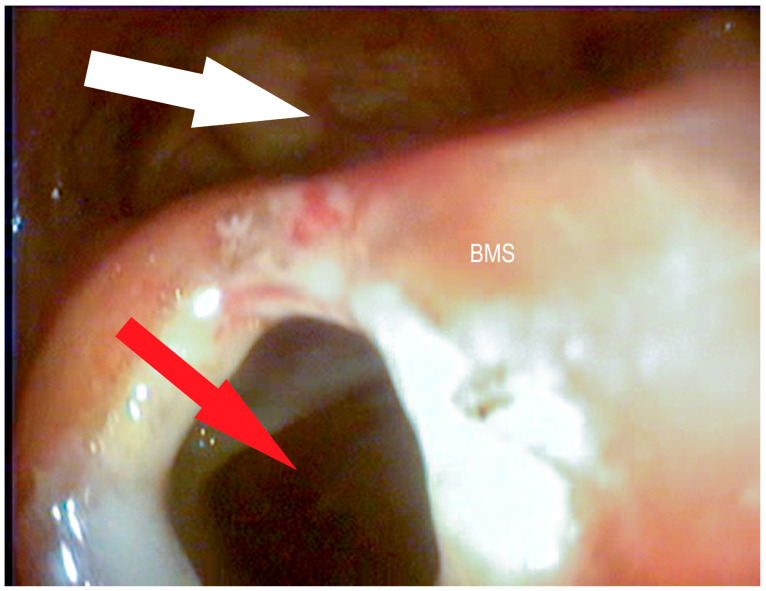
Bulla of the maxillary septum (BMS) after laserbullotomy; surgically created passage to the rostral maxillary sinus (red arrow); natural passage to the caudal maxillary sinus through the caudal sinonasal canal (white arrow) (Photo: Matthias Jehle, Vetmeduni Vienna).

**Figure 8 vetsci-09-00334-f008:**
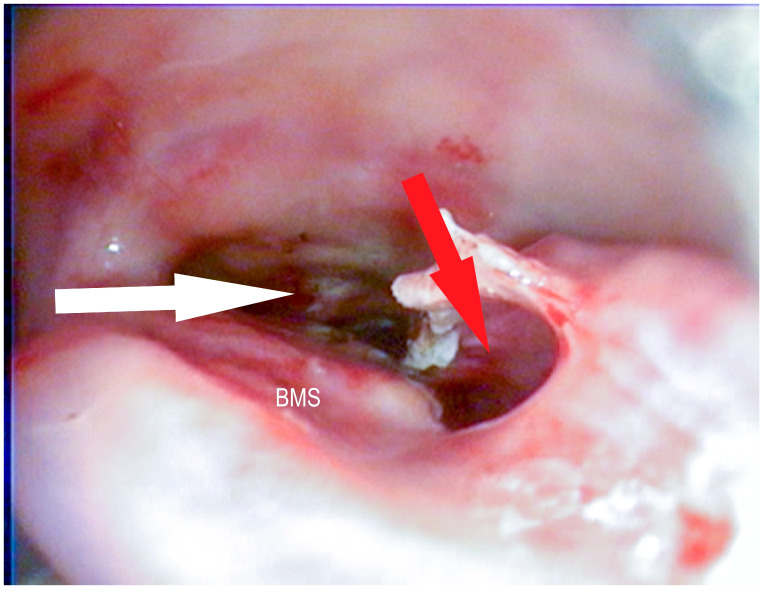
Bulla of the maxillary septum (BMS) after laserbullotomy; surgically created passage to the rostral maxillary sinus (red arrow); natural passage to the caudal maxillary sinus through the caudal sinonasal canal (white arrow) (Photo: Matthias Jehle, Vetmeduni Vienna).

**Figure 9 vetsci-09-00334-f009:**
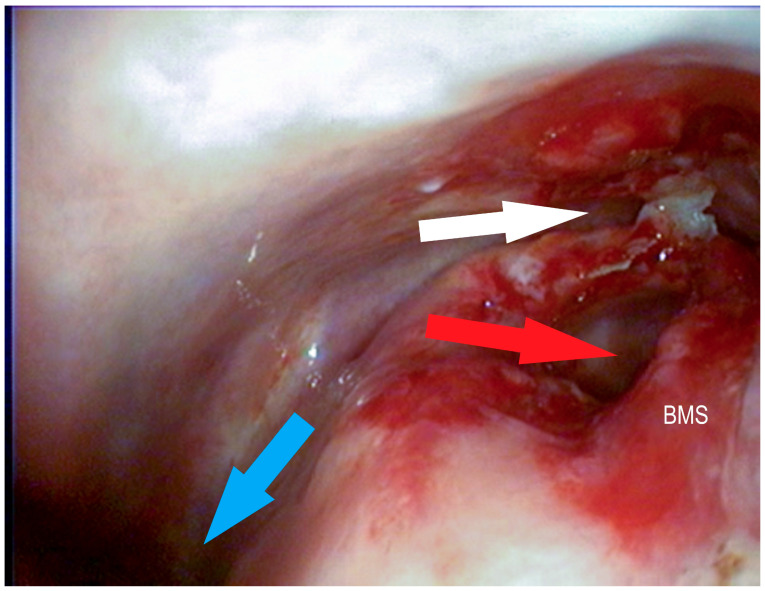
Bulla of the maxillary septum (BMS) after laserbullotomy; surgically created passage to the rostral maxillary sinus (red arrow); natural passage to the caudal maxillary sinus through the caudal sinonasal canal (white arrow) and to the common nasal meatus (blue arrow) (Photo: Matthias Jehle, Vetmeduni Vienna).

**Figure 10 vetsci-09-00334-f010:**
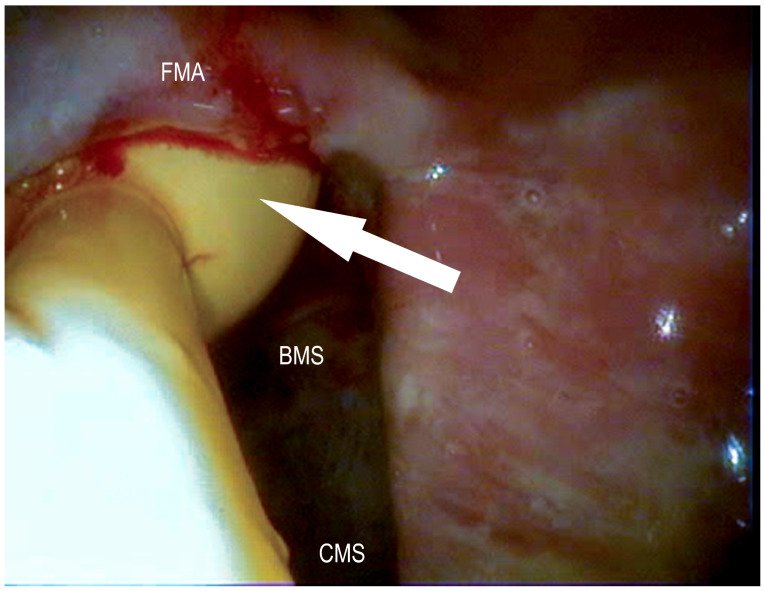
Foley catheter placed in a retrograde fashion in the caudal sinonasal canal (white arrow) after frontal trephination on top of the bulla of the maxillary septum (BMS); frontomaxillary aperture (FMA), caudal maxillary sinus (CMS) (Photo: Matthias Jehle, Vetmeduni Vienna).

**Table 1 vetsci-09-00334-t001:** Signalement, diagnosis, access to sinus and additional procedures for cases described in this study.

Patient	Age	Sex	Weight	Diagnosis	Exodontia	Access to Sinus	Additional Procedures	Complications/Additional Diagnoses
1	29	M	525	dental related sinusitis right CMS	110	NMA		MITSE 110; unspecified carcinoma CMS
2	23	M	434	dental related sinusitis right (RMS, VCS, CMC, CFS)	109	NMA	removal necrotic bone fragments from ventral conchae	necrosis ventral conchae
3	22	M	520	dental related sinusitis right RMS, VCS	108	NMA		
4	11	G	540	dental related sinusitis right RMS, VCS	108	NMA		
5	15	G	550	dental related sinusitis left RMS, VCS, CMS, CFS	208	NMA ± necrotic conchae	removal necrotic bone fragments from ventral conchae	necrosis ventral conchae
6	27	G	284	sinusitis left RMS, VCS, CMS, SCV due to orosinuidal fistula 210/211		NMA	removal necrotic bone fragments from ventral conchae	necrosis ventral conchae
7	16	G	542	dental related sinusitis right RMS, VCS;	109	NMA		
8	27	G	414	sinusitis left RMS, VCS, CMS, SCV due to orosinuidal fistula 210/211		NMA	removal necrotic bone fragments from ventral conchae	necrosis ventral conchae
				sinussitis right RMS, CMS due to multiple sinus cysts RMS		NMA		
9	19	M	595	sinusitis right RMS, VCS; granuloma roof ventral conchae/necrosis VC		NMA ± necrotic conchae	removal necrotic bone fragments from ventral conchae	necrosis ventral concahe
10	8	M	466	sinusitis left CMS due to fracture lacrimal and maxillary bone		NMA		fracture nasolacrimal duct
11	14	M	526	dental related sinusitis right RMS, VCS, CMS, CFS	109	NMA		
12	13	M		dental related sinusitis right RMS, CMS	109	Ballon Sinuplasty NMA		
13	10	M	458	sinusitis due to infected fracture right lacrimal, maxillary and frontal bone		Ballon Sinuplasty NMA	retrograde insertion of the catheter into caudal sinonasal channel from open fracture site in maxillary bone	septic suturitis lacrimal/maxillary suture;
				sinusitis due to infected fracture left lacrimal, maxillary and frontal bone		Laser fenestration BMS		septic suturitis lacrimal/maxillary suture
14	11	G	430	sinusitis RMS post fracture maxillary bone		Laser fenestration BMS		
15	24	M	572	dental related sinusitis left RMS, VCS	209	Laser fenestration BMS		
16	8	G	451	primary mycotic sinusitis right CMS, CFS		conchotomy medial wall dorsal chonchae/CFS		
						SENMAP		
17	33	G	365	dental related sinusitis and sinus cysts left RMS, VCS, CMS	209	conchotomy medial wall ventral chonchae/VCS		
						Trephination CFS	cyst removal and access to nasal cavity	
18	4	M	530	dental related sinusitis left RMS, CMS	209	conchotomy medial wall ventral chonchae/VCS		no passage over infraorbital canal to RMS-long reserve crowns 208/209
19	2	G	93	sinusitis due to developmental related deformation left sinus system and impairment of sinus drainage		SENMAP		
20	13	M	600	dental related sinusitis right RMS, VCS, CMS and sinus cysts RMS, VCS	108	SENMAP		
21	12	G		sinusitis right CMS, CFS post fracture nasal and frontal bone-adhesions in the area of caudal sinonasal channel-impairment sinus drainage		SENMAP		recurrent sinus empyema
22	19	M		dental related sinusitis left RMS, VCS	209	SENMAP		orosinoidal fistula post exodontia 209
23	19	M	633	ethmoid hematoma left and right VCS, RMS, CMS, CFS		bilateral SENMAP	tracheotomy; resection of REH remnants via diode laser and bipolar vessel sealing device	
24	20	M	700	dental related sinusitis and sinus cysts left VCS, RMS, CMS, CFS and	211	frontal bone flap left	laser resection cyst left RVC	SSI bone flap
				right RMS	108	maxillary trephination right		
25	26	G		sinusitis left VCS, RMS, CMS, CFS; orosinoidal fistula 209/210	210	SENMAP		

M: mare; G: gelding; VCS: ventral conchal sinus; RMS: rostral maxillary sinus; CMS: caudal maxillary sinus; CFS: conchofrontal sinus; NMA: nasomaxillary aperture; BMS: bulla of the maxillary septum; SENMAP: surgical enlargement of the nasomaxillary aperture.

## Data Availability

The data presented in this study are available on request from the corresponding author.

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
