# Peer review of "Trephination versus Minimally Invasive Transnasal Approaches for the Diagnosis and Treatment of Sinus Disease in Horses"

_vetsci, 2022, doi:10.3390/vetsci9070334_

Round 1
Reviewer 1 Report
The paper is well presented and interesting, the bibliography exhaustive.
The work can be presented as it is written, however I suggest some improvements.
Line 29-30 ... is uncommonly encountered.... this pathology is actually easily encountered, to write this sentence bring more bibliographic support.
Fig 9: ...no clinical relevance.... at the time of the endoscopy it does not matter ..... I would remove the sentence
In the discussion the advantages of traditional surgery compared to minimally invasive are little emphasized. I recommend to expand
Author Response
Dear valued reviewer,
please see the attachment.
Sincerly
Matthias Jehle

Reviewer 2 Report
Thank you very much for the manuscript on this very interesting and clinically relevant topic. It highlights the advantages and disadvantages of trephination versus minimal invasive transnasal approaches especially for treatment of sinus disease in horses. A treatment via invasive transnasal approaches seems to be preferred when a transnasal access to the affected sinus is possible via different technics. Though the cases should be selected carefully and considerable experience appears to be necessary. I am looking forward to studies with larger case numbers and long-term follow-up.
From my view there are only a few points to revise:
- Please consistent insert blank spaces between a rate and its unit
- In line 133 replace the comma of 1,2 mm by a period
- In line 150 between 5 mm and diameter the blank space is missing
Author Response

(The authors gave the same response as above.)

Reviewer 3 Report
This is an interesting paper that describes less invasive surgical approaches to the management of primary and secondary sinusitis in horses and that illustrates application of these approaches through a series of clinical cases. It is primarily a descriptive clinical report, which will be of interest to those in equine clinical practice.
The quality of the English language is often rather poor and the manuscript would benefit from editorial input from an individual with better command of English.
The authors might consider one or two simple line drawings to identify key anatomical landmarks and illustrate routes of passage of the endoscope in different procedures. These might be helpful to the reader.
I have a few relatively minor further comments:
Line Comment
7 “..aims to highlight..”
10-12 Please reword this sentence. (e.g. “…through the apertures..” – what apertures??)
16 It’s not he “clinical signs” that are treated but the disease responsible for the clinical signs
42 Do you mean “quantity” rather than “quality”?
43 Smell is another important clinical sign (e.g. associated with dental disease)
44 “thorough clinical exam”
48-52 I wonder of the following might capture what you are trying to say more clearly: “…or sinus systems, such as discharge from the drainage angle or mucosal swelling, have been reported as present in 96% of cases although endoscopy only contributes to a precise diagnosis in 21% of cases”.
69-70 “..surgical techniques, including maxillary and frontal trephinations and bone flaps, have..”
108 5.9mm (instead of 5,9)
109 length (not lengths)
133 1.2mm
140 “..enough to pass an endoscope through it”.
150 Are these power settings the correct way around?
Fig 7 Surgically created… (Same for other figures)
Tab 1 Inconsistent use of acronyms (e.g. VC or ventral conchae)
297 Creation of bone flaps was not mentioned in materials and methods
334 “complicated exodontia” is not a complication of the procedure per se
334 “carcinoma” is not a complication of the procedure
340 Please replace “patients” with “cases”
400 Do you mean “patience” rather than “patients” here?
Author Response

(The authors gave the same response as above.)
